# Selective Detection of Erythrocytes with QCMs—ABO Blood Group Typing [note 1]

**DOI:** 10.3390/s23177533

**Published:** 2023-08-30

**Authors:** Usman Latif, Alexandra Seifner, Franz L. Dickert

**Affiliations:** 1Department of Analytical Chemistry, University of Vienna, Waehringer Str. 38, A-1090 Vienna, Austria; usmanlatif@cuilahore.edu.pk (U.L.); alexandra.seifner@gmx.at (A.S.); 2Interdisciplinary Research Centre in Biomedical Materials (IRCBM), COMSATS University Islamabad, Lahore Campus, Lahore 54600, Pakistan

**Keywords:** blood group typing, molecular imprinting, quartz crystal microbalance, sensor, supra molecular chemistry

## Abstract

Blood transfusion, as well as organ transplantation, is only possible after prior blood group (BG) typing and crossmatching. The most important blood group system is that of Landsteiner’s ABO classification based on antigen presence on the erythrocyte surfaces. A mass sensitive QCM (quartz crystal microbalance) sensor for BG typing has been developed by utilizing molecular imprinting technology. Polyvinylpyrrolidone (crosslinked with N,N-methylenebisacrylamide) is a favorable coating that was imprinted with erythrocytes of different blood groups. In total, 10 MHz quartz sheets with two resonators, one for MIP (molecularly imprinted polymer) and the other for NIP (non-imprinted polymer) were fabricated and later used for mass-sensitive measurements. The structure of erythrocyte imprints resembles a donut, as identified by AFM (atomic force microscope). All the erythrocytes of the ABO system were chosen as templates and the responses to these selective coatings were evaluated against all blood groups. Each blood group can be characterized by the pattern of responses in an unambiguous way. The results for blood group O are remarkable given that all types of erythrocytes give nearly the same result. This can be easily understood as blood group O does not possess neither antigen A nor antigen B. The responses can be roughly related to the number of respective antigens on the erythrocyte surface. The imprints generate hollows, which are used for reversible recognition of the erythrocytes. This procedure is based on molecular recognition (based on supramolecular strategies), which results from size, shape and enthalpic interactions between host and guest molecules.

## 1. Introduction

A suitable crossmatch test between the intended donor and the patient is highly recommended for blood transfusion or organ transplantation because blood clump or agglutinate will occur in the event of mismatch, and will ultimately lead to serious consequences or sudden death. The ABO blood group (BG) system was first discovered by Austrian scientist, Karl Landsteiner, in 1900 and rhesus (Rh) BG later [1]. Today, successful and safe blood transfusion and organ transplantation is only possible because of his pioneering work. We can differentiate the red blood cells based on antigens present on their surfaces. The glycosyltransferase gene is responsible for the expression of ABO antigens on their surfaces [2,3]. Thus, blood group A contains only A antigens, B contains only B antigens, AB contains both A and B antigens, O does not have A or B antigens and Rh gives information about Rh antigens [4,5,6]. Blood group A is present in most of the population and can be categorized into prime subgroups as A_1_, A_2_ and other weak A subgroups [7,8]. The following blood groups were studied in this investigation as A_1_, A_2_, B, O, A_1_B and A_2_B (Table 1). The expression of A or B antigens is summarized in Table 1 [9,10,11]. Exact identification of these subgroups is important because these can be mistyped as O group individuals [12]. Ultimately, mismatched transfusion will lead to grave consequences.

Generally, the process of identifying the nature of antigens present on the surface of blood cells is called blood group (BG) typing [13]. Principally, BG typing refers to the distinct reaction between specific antibodies and antigens (present on erythrocyte surface) to monitor blood clamping or agglutination. There are a large number of methods for BG typing [14], which can be classified into classical, such as tube or slide tests, or modern methods [15], such as microplate or gel centrifugation. Some other modern-day methods utilize nucleic acid amplification, ultrasound back scattering [16], hybrid methods—laser radiation followed by digital imaging [17,18]—and polymerized chain reaction with sequence-specific priming (PCR-SSP) strategies for human BG typing with more accuracy [19]. Both classical and advanced methods aim at determination of rare or weak alleles of BG, but there is a compromise between sensitivity, analysis time and cost of that particular test. Moreover, highly trained personnel are required to interpret the test results of BG typing.

Here, a chemical sensor is proposed for determining blood groups. Instead of bulky instrumentations, miniaturized sensors offer on-spot measurements with required sensitivity, selectivity and relatively less response time [20,21]. In this study, a mass-sensitive transducer was combined with a coating for molecular recognition. Robust QCM resonators were applied, which generate frequency responses to mass load. Selectivity was achieved using MIP coatings [22,23,24], which selectively enrich the respective BG at the QCM surface. Thus, a supramolecular strategy for erythrocyte recognition was designed.

In this study, blood group typing has been performed by combining molecular imprinting technique and QCM [25,26]. The use of MIPs allows us to recognize erythrocytes while interacting with the whole cell surface. Moreover, the results correlated to the number of erythrocyte antigens in a semi-quantitative way. Furthermore, the erythrocytes were detected under isotonic conditions to avoid cell damaging.

## 2. Materials and Methods

### 2.1. Reagents

The monomer 1-vinyl-2-pyrrolidone was purchased from Merck (Rahway, NJ. USA), whereas N,N-methylenebisacrylamide as crosslinker and azobisisobutyronitrile (AIBN) as initiator were purchased from Fluka (Charlotte, NC, USA). Whole blood samples were provided by Blood Donation Center of the Austrian Red Cross (Vienna, Austria).

### 2.2. Instrumentation

The 10 MHz QCM discs (Great Microtama Electronics, Surabaya, Indonesia) in a diameter of 14 mm were used for screen printing two gold electrodes by using gold paste (Heraeus, Hanau, Germany) in a diameter of 4.5 mm. These electrodes are very robust. These two electrodes on quartz sheet were used to compensate for nonselective effects or temperature fluctuations by differential measurements. The mass-sensitive sensor was placed in a custom-made cell made from PDMS, which can accommodate a sample volume of 50 µL. The measuring cell connected to a frequency counter (Agilent 53131A, Agilent Technologies, Santa Clara, CA, USA) via a custom-made oscillating circuit. The electrode pairs on the quartz were part of the feedback loop of the oscillator as frequency-determining elements. Any load on the oscillator output can damp and detune the oscillator frequency. The resonance frequency of QCM will decrease proportionally to the mass load, according to the Sauerbrey equation [27,28]. As counteraction, a buffer amplifier was used to decouple the oscillator output from the following stage. The whole circuit board included two oscillators for measuring and for the reference channel. Additionally, a power supply was also integrated. In this way, differential measurements were performed to eliminate temperature and viscosity effects. Thus, the responses of MIP and NIP coatings could be compared. The results were then transferred to computer screen by using LabView software, which measured the reading after every 6 s. The sample solutions were exposed to a mass-sensitive sensor by using a peristaltic pump driven at 0.5 mLmin^−1^. An atomic force microscope (Veeco Nanoscope Iva, Plainview, NY, USA) was utilized for surface profiling in contact mode. The goal of this paper was blood group typing. Additionally, the erythrocytes were characterized by AFM to avoid cell damage. Moreover, the essential criterion while designing QCM-based sensor was its quick response, as well as fast recovery time, which are discussed in the next sections.

### 2.3. Imprinting of Polyvinylpyrrolidone Layers by Erythrocytes

In molecular imprinting, exact mapping of analytes is necessary during curing of the monomer mixture for successful recognition of the templated analyte. The selection of the polymer system and the quantitative ratio of the monomers used is, thus, of great importance. The hydrophilic nature of polyvinylpyrrolidone makes it a favorable candidate in developing a sensor system for erythrocyte detection in aqueous media. This polymer forms efficient hydrogen bonds, which is also proven by its solubility in water. Polymerization takes place via radicals generated by the initiator, azobisisobutyronitrile (AIBN).

The vinylpyrrolidone monomer mixture was prepared from 95 mg of 1-vinyl-2-pyrrolidone, 5 mg N,N-methylenebisacrylamide as crosslinker and 1 mg azo-bis-(isobutyronitrile) (AIBN). Solvent for the crosslinker was 1-vinyl-2-pyrrolidone itself. The 5 mg crosslinkers in 95 mg 1-vinyl-2-pyrrolidone must be dissolved at 70 °C. At this temperature, 5 mg corresponds approximately to the solubility limit of the crosslinker in vinylpyrrolidone. About 1 mg AIBN was added stepwise (at this step, care must be taken while adding AIBN because the large volume will polymerize the vinylpyrrolidone quickly). This mixture can either be exposed to UV light or 70 °C until pre-polymerization. Figure 1 shows the AFM image of the imprint, corresponding to the diameter of intact blood cells with a size of about 7–8 μm and depth of about 200 nm. The blood cells seemed to maintain their geometric shape, especially that of a donut [29]. The imprinting process was performed by sedimentation of erythrocytes; thus, the template was engulfed by the prepolymer, as shown in Figure 2. For this purpose, 2 µL of pre-polymer was dispensed on a gold electrode of QCM, and afterwards, 1 µL of freshly washed erythrocytes was exposed to this thin pre-polymer layer. It was then exposed to UV light for about 3 h for rapid polymerization and then kept at room temperature for 2 days until complete polymerization. The depth of the imprints is of great importance. If imprints are deep into the polymer, then these would not be accessible to the analyte. Moreover, washing is an important step because it should not only remove the template, but the geometric shape should be retained. For this purpose, isotonic NaCl solution (which does not damage blood cells) was used for removing blood cells by washing, while 10^−3^ Vol% of Triton-X was used to remove protein contaminants. However, the erythrocytes that are engulfed on the surface can be removed easily in comparison to those who are deeply buried inside the polymer. Thus, washing off loosely bound protein as well as templated analyte ensures an effective reinclusion of erythrocytes.

## 3. Results and Discussion

The erythrocytes of blood group A_1_ were washed with isotonic NaCl solution before imprinting and mass-sensitive measurements. Approximately 100 μL of blood was mixed with one mL of isotonic NaCl solution. Further improvements were achieved by adding the nonionic surfactant Triton X-100 to the isotonic NaCl solution in a concentration of 10^−3^ vol %. The suspension was mixed well and centrifuged at 1200 rpm for approximately 7 min. After centrifugation, the erythrocytes were deposited at the bottom. The lighter components, such as hemoglobin, low-molecular components in plasma, remained in the supernatant, which was removed. The process was repeated until the solution became almost colorless. The resulting erythrocyte suspension at the bottom was used for the imprinting process. This procedure was used both for the imprinting of erythrocytes and the analytes. Thus, damaged erythrocytes were removed if degraded blood samples were tested.

The A_1_-erythrocyte imprinted sensor was exposed to its templated analyte as well as the other erythrocyte subgroups, A_2_, A_1_B, A_2_B, O and B, and the results are shown in Figure 3. The sensor depicts the highest response towards its templated analyte A_1_, whereas highest cross-selectivity was observed towards to A_1_B due to the presence of a high number of A antigens. Blood group A_1_ contains a higher number of A antigens on the surface in comparison to A_2_ or B groups. Highest cross-selectivity is caused by A_1_B, followed by B and A_2_B. A_1_B subtype contains a higher number of A antigens in comparison to A_2_B, which is why it shows higher cross-selectivity. The sensor responses towards other blood groups, i.e., A_2_ and O are somewhat very less in comparison to other blood groups [30].

In the next step, an A_2_-imprinted sensor was exposed to templated and other analytes and the responses are shown in Figure 4.

The measurements show that the highest frequency shifts of the sensor result towards templated analyte A_2_. The highest cross-selectivity was observed towards A_1_B, followed by A_2_B and then A_1_. On the other hand, very low frequency responses were observed in the case of B and O blood groups, given that these blood groups possess no A antigens. The type of A antigens on the surfaces of subgroups A_1_, A_2_, A_1_B and A_2_B are similar and the only differentiation lies in the number of antigens. These results show that the whole surface structure of the erythrocyte seems to be important and not only the number of antigens. These results reveal successful imprinting effects. The surface structure of the erythrocytes varies with the change in the number of antigens. Subtype A_2_ has a lower number of antigens in comparison to A_1_, but it shows higher sensor response. The MIP sensor layer contains all A_2_ complementary sites on their surface, which successfully recognize its templated analyte. Even though the types of antigens are the same, variation in their number changes the overall erythrocyte structure and, ultimately, the imprinting sites.

The concentration dependence responses of the A_1_B-imprinted sensor are shown in Figure 5. The sensor’s performance is nearly linear up to concentrations of 6 × 10^8^ erythrocytes/mL. Then, saturation effects were observed. The subtype A_1_B erythrocyte contains both A and B antigens on its surface. Imprinting with subtype A_1_B will create optimized recognition sites that are complementary to A_1_B erythrocytes, but it also shows some affinity to subtypes of blood group A and B, too. Blood groups A_1_ and A_2_ show approximately half of the A_1_B response. The A_1_B printed polymer provides A_1_ cavities for the inclusion of A_1_ antigens of blood group A. Erythrocyte A_1_B contains a greater number of A antigens than B antigens. Some additional affinity exists for antigen A in respect to B imprints, which is not shown in Figure 5. It is especially remarkable, however, that A_1_ blood does not give a similar sensor response as for blood group A_1_B, given that A_1_ includes a higher number of antigens A than in A_1_B. Obviously, the whole surface structure plays an important role, which is characterized by the imprinting process.

The responses in Figure 6 indicate that the corresponding sensor characteristic shows a linear behavior for blood group B, given that no saturation effects were observed. The response of A_2_ to B-imprinted coatings is nearly negligible in contrast to B, proving a successful imprinting procedure with the template. The minor effect of blood A_2_ group can be easily understood due to the absence of B antigens. In contrast to this, A_1_B yields a more than twofold response to blood group B as it contains an appreciable amount of B antigens.

In the next step, the polymer was imprinted with blood type O. Blood group A contains A antigens, B contains B antigens, AB contains both A and B antigens, whereas O blood group does not have either A or B antigens. That is why the O-BG-imprinted sensor responses to all blood groups are quite similar, as shown for erythrocytes A_1_, A_2_, A_1_B, A_2_B and B for the distinct blood groups in Figure 7. Responses of NIP are included in addition to those of MIP in this figure. The NIP responses were nearly negligible in comparison to MIP; thus, they were not shown in other figures.

Figure 8 shows that the most pronounced response is always obtained if the analyte is identical to the template. The highest cross-sensitivity for template A_1_ was obtained for blood group A_1_B because of its high number of A antigens (Table 1). Remarkable are the findings for templating with A_2_. The template A_2_ yields the highest response, although erythrocytes A_1_ and A_1_B possess a higher number of A antigens. Printing with A_1_B causes some cross-sensitivities to A_1_, A_2_ and A_2_B, as these blood groups contain an appreciable number of A_1_ antigens. The template A_2_B generates cross-sensitivity to A_1_B and B, as these blood groups contain an appreciable number of B antigens. Cross-sensitivities for B templating are caused by blood group A_2_B due to their B antigens. Appreciable cross-sensitivities are also generated by blood group O, obviously caused by their high number of H antigens. A characteristic selectivity for different blood groups was not observed while imprinting with blood group O. These findings are obvious as O-BG contains neither A nor B antigens.

The selectivity scheme can be understood through principles of supramolecular chemistry, which are realized by molecular imprinting in this case [31]. Thus, the printing process generates patterned polymers for reinclusion of antigens. It is obvious that A_2_ imprinting will generate interaction sites for antigen A stemming from blood group A_1_. Thus, it could be expected that typing by A_2_ will create the same sensor response for blood group A_1_ and A_2_, given that A_1_ contains more antigens A than A_2_. Thus, it could be concluded that imprints will bind A antigens of A_1_, too. However, the findings show that printing with erythrocytes of A_2_ dominates the A_2_ detection in comparison to A_1_. Generally, imprinting with a distinct blood group always leads to an outstanding sensor response for the same blood group. This could be understood by assuming a chemical argument for steric hindrances, which do not allow a tight binding to the sites of imprints [29]. Obviously, the complex surface composition of erythrocytes prevents an efficient binding of antigens if the printing blood group and analyte blood group are not identical. Generally, the MIPs characterize the whole surface and not only an epitope. This phenomenon was also observed for detecting viruses with MIP sensors [32].

The strategy (washing with isotonic NaCl solution) used to measure erythrocytes guaranteed that only undamaged cells were tested. Blood samples show some degradation phenomena if whole blood in an adequate concentration is the basis of measurements. Then, over the course of time, the sensor responses increase due to the assembling of aggregates on the QCM electrodes. The blood sample gets degraded after 9 months, thus, the characteristic selectivity pattern vanishes. The reason for degradation is the complex structure with a highly organized membrane with many functions. Osmotic effects must be especially emphasized. The sensor layers consist of cross-linked polymers with polar properties. Thus, these coatings are much more stable than erythrocyte cells. The selectivity pattern does not alter for several weeks for the same sensor. The sensor can also be fabricated in a reproducible way.

## 4. Conclusions

Erythrocytes of human ABO blood groups, as well as their subtypes, were used as analytes to develop a sensor system for blood typing. The strategy used can be considered as supramolecular chemistry, which considers both intermolecular interactions and shape recognition. This can be realized by molecular imprinting without tedious synthetic work. Chemical, as well as structural, properties of the analytes were transferred into a polymer layer with the help of a molecular imprinting technique. This means that the template is applied to a prepolymer, where self-assembling of monomer components around erythrocytes leads to an embedding due to weak chemical interactions. In total, 10 MHz of quartz crystal microbalances (QCMs) were used as transducer devices. Erythrocyte imprints in polymer coatings were clearly observed via an atomic force microscope. The vinylpyrrolidone/N,N-Methylenebisacrylamide copolymer system was used for erythrocyte imprinting, given that the monomers are soluble in water. Thus, polarity was adapted to that of erythrocytes. The selectivity of the imprinted layers was investigated by exposing these sensors to blood groups of the ABO system, such as A_1_, A_2_, A_1_B, A_2_B, B and O. Mass-sensitive sensors were used to enrich these erythrocytes at the coating surface according to an optimized chemical recognition. All the sensors that were fabricated in this study are reusable. For example, a sensor that was designed (templated with A_1_) to measure analyte A_1_, the same sensor was exposed to other interfering species, such as A_2_, A_1_B, A_2_B and O. In every case, a pronounced response was obtained if the analyte was identical to the template. The sensitivity pattern obtained can be correlated to the number of antigens on the erythrocyte surface. Furthermore, the whole surface structure must be considered, which may lead to steric effects explaining subgroup selectivity. Thus, this MIP sensor shows biochemical recognition exceeding those of a few distinct sites of epitopes, but whole surface structures in nm to µm dimensions can be characterized. This article expresses that blood group typing is possible by coating MIPs as a recognition layer on QCMs (transducer). No paper that has performed ABO blood group typing by combining MIPs with QCMs in such a way is known to us. Moreover, sensor responses correlated with erythrocyte antigens in a semi-quantitative way.

## Figures and Tables

**Figure 1 sensors-23-07533-f001:**
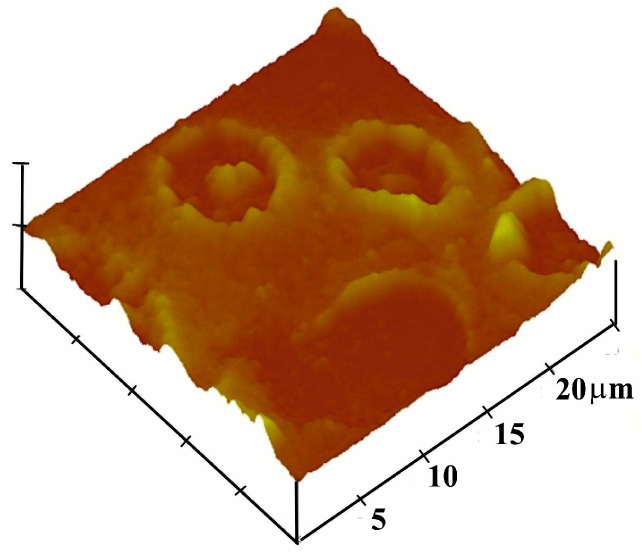
AFM image of complete erythrocyte imprints (forming a donut shape) on polyvinylpyrrolidone film.

**Figure 2 sensors-23-07533-f002:**
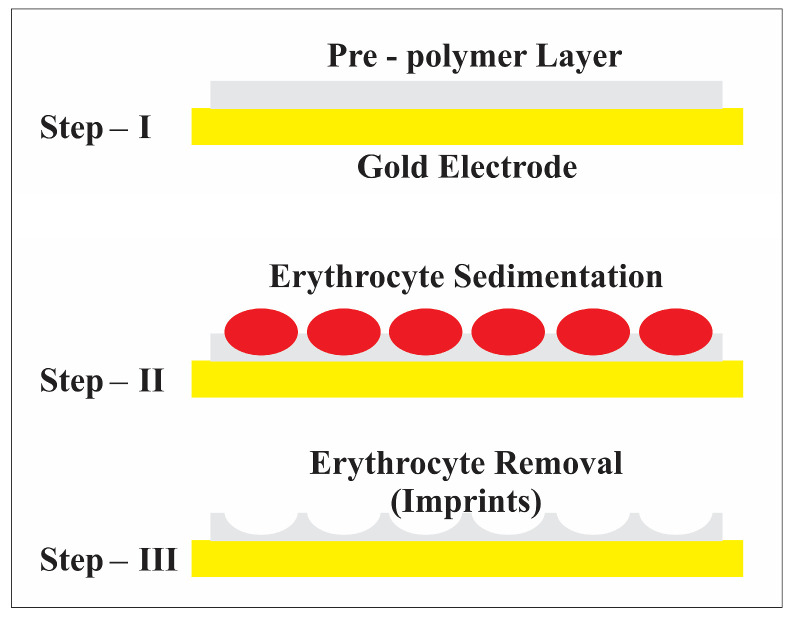
Schematic representation of the erythrocyte imprinting process.

**Figure 3 sensors-23-07533-f003:**
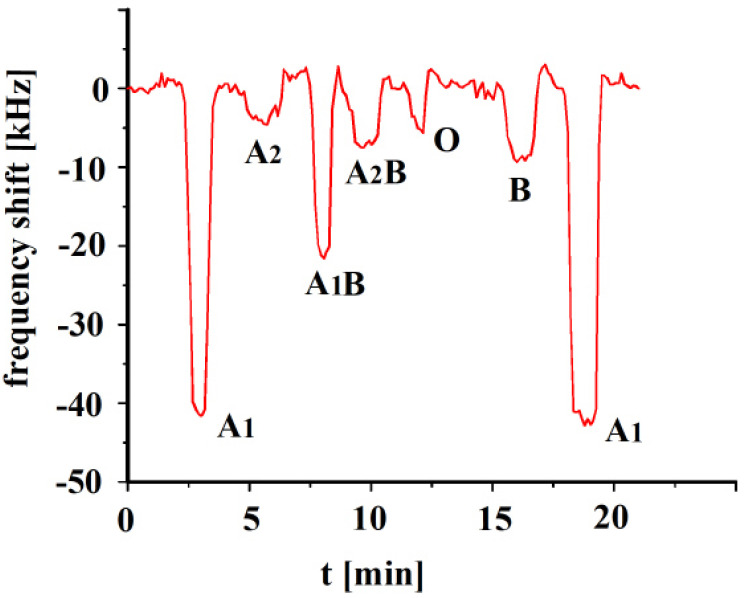
Polyvinylpyrrolidone/N,N-methylenebisacrylamide sensor layer printed with A_1_-erythrocytes, both imprint and analytes were washed with isotonic NaCl and 10^−3^ vol. % Triton X-100 surfactant, 10 MHz QCM sensor frequency responses to 3.5 × 10^8^ erythrocytes/mL as a function of time to different blood groups.

**Figure 4 sensors-23-07533-f004:**
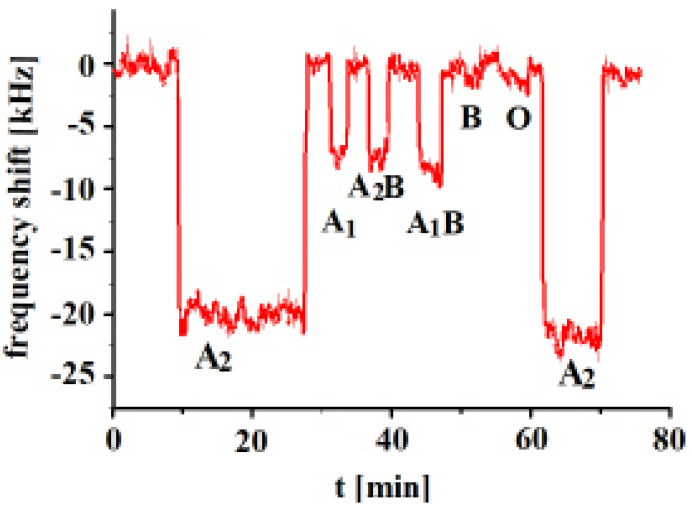
Polyvinylpyrrolidone/N,N-Methylenebisacrylamide sensor layer printed with A_2_-erythrocytes, 10 MHz QCM sensor frequency responses to 3.5 × 10^8^ erythrocytes/mL as a function of time to different blood groups.

**Figure 5 sensors-23-07533-f005:**
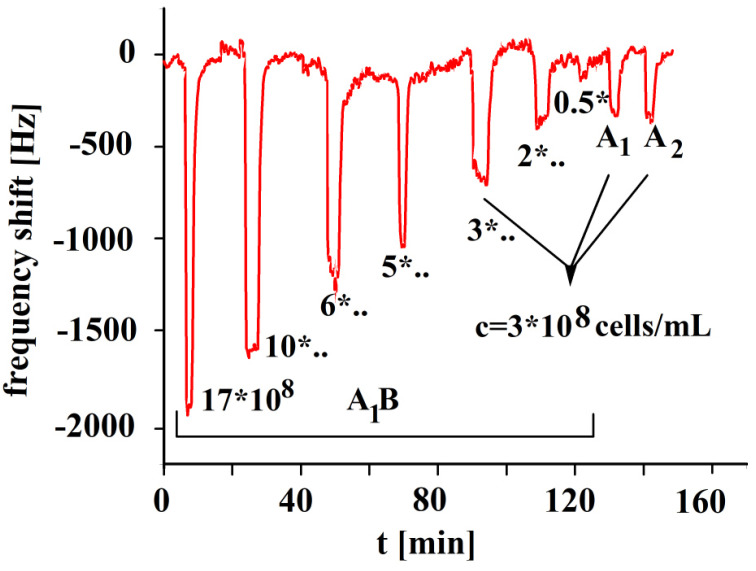
Polyvinylpyrrolidone/N,N-Methylenebisacrylamide sensor layer printed with A_1_B-erythrocytes, 10 MHz QCM sensor frequency responses to the template at varying erythrocyte concentrations (17, 10, 6, 5, 3, 2, 0.5 × 10^8^ cells/mL) and A_1_ and A_2_ blood groups as a function of time.

**Figure 6 sensors-23-07533-f006:**
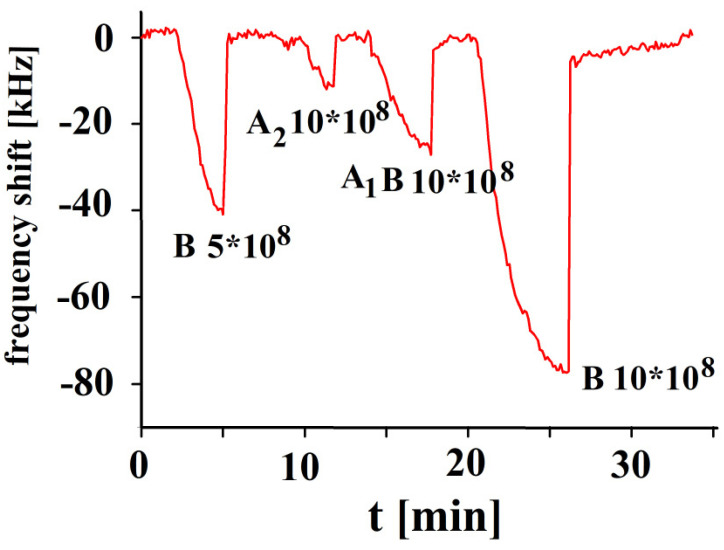
Polyvinylpyrrolidone/N,N-Methylenebisacrylamide sensor layer printed with B-erythrocytes, 10 MHz QCM sensor frequency responses to different concentrations of erythrocytes as function of time to blood groups B, A_2_ and A_1_B in concentrations of 10 × 10^8^, and 5 × 10^8^ erythrocytes/mL.

**Figure 7 sensors-23-07533-f007:**
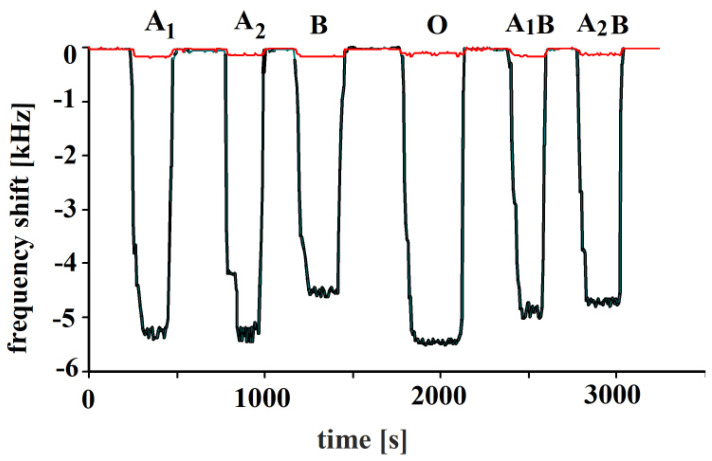
Polymerized vinylpyrrolidone/N,N-Methylenebisacrylamide sensor layer was printed by blood group O erythrocytes, 10 MHz QCM sensor responses to erythrocytes to all other blood groups in concentrations of 10^8^ cells/mL are shown. Minor responses are due to the NIP layers.

**Figure 8 sensors-23-07533-f008:**
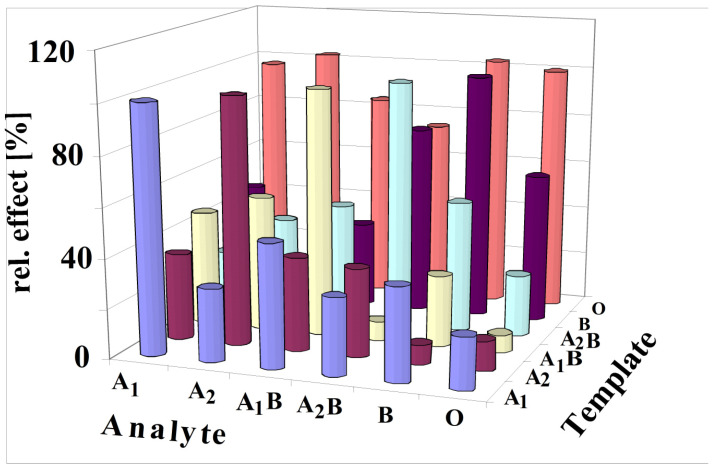
All results of imprinted vinylpyrrolidone/N,N-Methylenebisacrylamide copolymers with all antigens of the ABO system are summarized in this figure (3.5 × 10^8^ erythrocytes/mL). The template analyte always yields the most pronounced effect. The responses for each template are calibrated to the maximum response as 100%.

**Table 1 sensors-23-07533-t001:** Number of antigens A and B on erythrocytes in the ABO blood group system.

Blood Group	A-Antigen	B-Antigen	H-Antigen
A_1_	810,000–1,700,000		70,000–170,000
A_2_	160,000–440,000		More anti-H activity
B		610,000–830,000	400,000–470,000
O			1,600,000–1,900,000
A_1_B	420,000–850,000	310,000–560,000	
A_2_B	120,000	310,000–500,000	

## Data Availability

Not applicable.

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
