# Peer review of "Selective Detection of Erythrocytes with QCMs—ABO Blood Group Typing†"

_sensors, 2023, doi:10.3390/s23177533_

Round 1

Reviewer 1 Report

This study has no novelty in the literature. The English is very poor and the manuscript should be completely re-written, reviewed by a native English-speaking person and submitted to a different Journal. Therefore, I recommend "reject".

Author Response

thanks!

Reviewer 2 Report

In this study, the authors applied molecular imprinting technology to a mass-sensitive transducer (quartz crystal microbalance, QCM) for blood typing. Erythrocytes were used as templates for patterning polymer surfaces. Due to the differential affinity of molecularly imprinted polymers towards template species and other species, QCM sensors enabled the recognition of different ABO blood groups. This study is expected to capture the interest of readers, as it presents a promising supramolecular approach for blood typing. However, some issues have to be solved before publication.

1.      It is recommended to add a relevant description of how the layer materials are integrated with the QCM sensors in the experimental methods section.

2.      When abbreviations such as QCM, MIP, NIP are first mentioned, their full forms should be provided.

3.      It is suggested to supplement relevant studies on the stability and reproducibility of the sensor.

4.      It is recommended to specify the analyte concentrations in Figure 7.

5.      The usage of "Figure" or "Fig." should be consistent throughout the text.

Author Response

Reviewer_2

In this study, the authors applied molecular imprinting technology to a mass-sensitive transducer (quartz crystal microbalance, QCM) for blood typing. Erythrocytes were used as templates for patterning polymer surfaces. Due to the differential affinity of molecularly imprinted polymers towards template species and other species, QCM sensors enabled the recognition of different ABO blood groups. This study is expected to capture the interest of readers, as it presents a promising supramolecular approach for blood typing. However, some issues have to be solved before publication.

  1. It is recommended to add a relevant description of how the layer materials are integrated with the QCM sensors in the experimental methods section.

            The imprinting process was performed by sedimentation of erythrocytes, thus the template is engulfed by the prepolymer as shown in figure 2. For this purpose, 2 µL of pre-polymer was dispensed on gold electrode of QCM and afterwards, 1 µL of freshly washed erythrocytes were exposed to this thin pre-polymer layer. It was then exposed to UV light for about 3 hours for rapid polymerization and then, kept it at room temperature for 2 days until complete polymerization. (Pg 3, lines 113-118)

  1. When abbreviations such as QCM, MIP, NIP are first mentioned, their full forms should be provided.

The full forms of different abbreviations have now been provided at the start.

  1. It is suggested to supplement relevant studies on the stability and reproducibility of the sensor.

            The fabricated sensor has been utilized to measure analytes multiple times and different sensors have been fabricated on similar principle which confirms that geometrical features of the analyte have been successfully transferred to the sensor via molecular imprinting. The sensor layer had a much higher stability than blood samples, many weeks and months. The coworker tested it, but also successors at a later date were able to do this, within better than 10% in reproducing the results.

  1. It is recommended to specify the analyte concentrations in Figure 7.

The analyte concentrations in figure 7 (now figure 8) is 3.5*108erythrocytes/mL of blood

  1. The usage of "Figure" or "Fig." should be consistent throughout the text.

            It has been corrected and now its consistent throughout the manuscript.  

Reviewer 3 Report

This study presents the detection of erythrocytes with MIP and NIP coated QCM transducers for blood group typing.  QCM-based bio-chemical sensors have been studied for a very long time and a lot of work has been done on them. Blood group typing is interesting. Some minor revisions and editing are suggested:

  1. The authors previously published papers about QCM based blood group typing. Please explain what is the new in this manuscript.
  2. Please clarify the device fabrication and measurement details in experimental section. A schematic diagram should improve clarity for the reader.
  3. In the perspective of selectivity, if you use a blood of an un-known person how you decide blood group of this person. As you know, the concentration of erythrocytes in 1ml of blood varies according to gender and age. Please explain and discuss.
  4. Are these sensors disposable or reusable? If they are disposable, it is not cost-effective. If they are reusable, how do you clean the sensors? Please explain
  5. Please explain the sensing mechanism of the sensor device clearly.
  6. The authors wrote that “The sensor can also be fabricated in a reproducible way.” How do you test the reproducibility? Please clarify.

Author Response

Reviewer 3

This study presents the detection of erythrocytes with MIP and NIP coated QCM transducers for blood group typing. QCM-based bio-chemical sensors have been studied for a very long time and a lot of work has been done on them. Blood group typing is interesting. Some minor revisions and editing are suggested:

  1. The authors previously published papers about QCM based blood group typing. Please explain what is the new in this manuscript.

This paper highlights the detection of all blood groups of the ABO system, including sub-groups A1B, A2B. Exact identification of these subgroups are important because these can be mistyped as O group which can lead to serious consequences. This was possible by a blood donation of the Vienna blood center.

  1. Please clarify the device fabrication and measurement details in experimental section. A schematic diagram should improve clarity for the reader.

The imprinting process was performed by sedimentation of erythrocytes, thus the template is engulfed by the prepolymer as shown in figure 2. For this purpose, 2 µL of pre-polymer was dispensed on gold electrode of QCM and afterwards, 1 µL of freshly washed erythrocytes were exposed to this thin pre-polymer layer. It was then exposed to UV light for about 3 hours for rapid polymerization and then, kept it at room temperature for 2 days until complete polymerization. (Pg 3, Lines 113-118)

  1. In the perspective of selectivity, if you use a blood of an un-known person how you decide blood group of this person. As you know, the concentration of erythrocytes in 1ml of blood varies according to gender and age. Please explain and discuss.

The measurements have shown, that a distinct amount of erythrocytes will lead to greatest response if analyte and template are identical! Gender variation were not studied.

  1. Are these sensors disposable or reusable? If they are disposable, it is not cost-effective. If they are reusable, how do you clean the sensors? Please explain

All the sensors which fabricated in this study are reusable. For example a sensor which was designed (templated with A1) to measure analyte A1, the same sensor was exposed to other interfering species such as A2, A1B, A2B, O etc. (Pg 9, lines 306-308). The sensors can be regenerated as shown in the figures.

  1. Please explain the sensing mechanism of the sensor device clearly.

The electrode pairs on the quartz are part of the feedback loop of the oscillator as frequency-determining element. Any load on the oscillator output can damp and detune the oscillator frequency. The resonance frequency of QCM will decrease proportionally to the mass load, according to the Sauerbrey equation (Latif, U.; Can, S.; Hayden, O.; Grillberger, P.; Dickert, F.L. Sauerbrey and anti-Sauerbrey behavioral studies in qcm sensors—detection of bioanalytes. Sensors and Actuators B: Chemical 2013, 176, 825-830.) and (Dickert, F.L.; Latif, U. 2.10 - quartz crystal microbalances: Chemical applications. In Comprehensive supramolecular chemistry ii, Atwood, J.L., Ed. Elsevier: Oxford, 2017; pp 201-211.) (Pg 2, lines 82-85)

  1. The authors wrote that “The sensor can also be fabricated in a reproducible way.” How do you test the reproducibility? Please clarify

The coworker tested it, but also successors at a later date were able to do this, within better than 10% in reproducing the results.

Reviewer 4 Report

I think this is an excellent contribution to celebrate Prof Lieberzeit's 50th birthday. I have a few minor queries/suggestions.

-Please add error bars where appropriate (such as Figure 7).

-With QCM, it is my understanding that you look at different overtones. The reported frequency shift, at which overtone is this?

-The novelty is not very clear from the introduction and there are other MIP sensors that focus on bloodtyping. Please compare your result to those. 

-Comment on how the QCM can be used as a portable system, cost and measurement time would need to be considered.

-Have the results been validated against other methods?

Author Response

Reviewer 4:

I think this is an excellent contribution to celebrate Prof. Lieberzeit's 50th birthday. I have a few minor queries/suggestions.

  1. Please add error bars where appropriate (such as Figure 7).

            We have shown relative effect in percentage of 6 different sensors while exposing to 6 different analytes. It’s the relative percentage of readings (average of triplicate) of each sensor and combine into one graph (now figure 8), error was less than 10 %. 

  1. With QCM, it is my understanding that you look at different overtones. The reported frequency shift, at which overtone is this?

We used a home-made oscillator circuit for measurements of frequency responses. An optimized signal is obtained in this case, if the oscillator is tuned to 10 MHz in this case. Its electronic bandwidth will cut off overtones and additional noise.

  1. The novelty is not very clear from the introduction and there are other MIP sensors that focus on blood typing. Please compare your result to those.

This paper highlights the detection of all blood groups of the ABO system in in a semi-quantitative analysis, but sub-groups A1B, A2B were also measured. This was possible by a blood donation of the Vienna blood center. Exact identification of these subgroups are important because these can be mistyped as O group which can lead to serious consequences.

  1. Comment on how the QCM can be used as a portable system, cost and measurement time would need to be considered.

Have the results been validated against other methods?

We have got blood from Vienna red cross, which was tested, there.

A general comparison (on the basis of principle, their use, cost as well as measurement time) among some of the classical and modern strategies on blood group typing is already given in our previous study i.e.; (Mujahid, A.; Dickert, F. L., Blood Group Typing: From Classical Strategies to the Application of Synthetic Antibodies Generated by Molecular Imprinting. Sensors 2016, 16, (1), 51.)

Round 2

Reviewer 1 Report

The subject is interesting and the results are presented in scientific way. I recommend its publication with following suggestions. 

1.     What is its novality, when you compare this study with the literature?

2.     The manuscript needs more updated references to extended the descriptions and supported the results. The manuscript has several typing and grammatical errors and requires improving English. 

3.     In the introduction, the authors need to discuss the use of quartz crystal microbalance (QCM) in the different studies. The reader should be explained in detail the principle of QCM. QCM studies should be explained in detail by referring to the sample articles below, https://doi.org/10.1016/j.msec.2018.12.086,https://doi.org/10.3390/bios12060371, https://doi.org/10.1016/j.ab.2022.114981,

4.     The purpose of this review has not been fully expressed to the reader.

5.     ‘Characterization studies’ part needs more detailed information about the analysis.

6.     What is the difference or novelty of this study from the reviews written on this subject in the literature?

Only minor English errors can be corrected in the article.

Author Response

Dear Reviewer,

I am sorry I had a blind eye and did not see your valuable suggestions,

F.L. Dickert

The subject is interesting and the results are presented in scientific way. I recommend its publication with following suggestions. 

  1. What is its novelty, when you compare this study with the literature?

In this study, blood group typing has been performed by combining molecular imprinting technique and QCM. The use of MIPs allows us to recognize erythrocytes while interacting whole cell surface. Moreover, the results were correlated to the number of erythrocyte antigens in a semi-quantitative way. Furthermore, the erythrocytes were detected under isotonic conditions to avoid cell damaging. (Pg 2, lines 70-74)  

  1. The manuscript needs more updated references to extended the descriptions and supported the results. The manuscript has several typing and grammatical errors and requires improving English.

Two more papers are cited as suggested by the reviewer (References 25 & 26). Moreover, the typing as well as grammatical errors in the manuscript have been corrected. 

3        In the introduction, the authors need to discuss the use of quartz crystal microbalance (QCM) in the different studies. The reader should be explained in detail the principle of QCM. QCM studies should be explained in detail by referring to the sample articles below, https://doi.org/10.1016/j.msec.2018.12.086,https://doi.org/10.3390/bios12060371, https://doi.org/10.1016/j.ab.2022.114981,

The papers of Denzli and  Skliar are now cited as suggested by the reviewer. These papers show similarities to our papers in respect to the methodology.

We used 10 MHz QCM discs in a diameter of 14 mm (Great Microtama Electronics Indonesia) were used for screen printing two gold electrodes by using gold paste (Hanau – Germany, Degussa) in a diameter of 4.5 mm. These electrodes are very robust. These two electrodes on quartz sheet were used to compensate for nonselective effects or temperature fluctuations by differential measurements. (Pg 2, lines 82-86)    

  1. The purpose of this review has not been fully expressed to the reader.

This article expresses that blood group typing is possible by coating MIPs as recognition layer on QCM (transducer). (Pg 10, lines 324-325)

  1. ‘Characterization studies’ part needs more detailed information about the analysis.

The goal of this paper is blood group typing. Additionally, the erythrocytes were characterized by AFM to avoid cell damaging. Moreover, the essential criterion while designing QCM based sensor is its quick response as well as fast recovery time which are discussed in the next sections. (Pg 3, lines 101-104)

  1. What is the difference or novelty of this study from the reviews written on this subject in the literature?

No paper is known to us which performs ABO blood group typing by combining MIPs with QCMs in such a way. Moreover, sensor responses are correlated with erythrocyte antigens in a semi-quantitative way. (Pg 10, lines 325-328)  

Comments on the Quality of English Language….Only minor English errors can be corrected in the article.

These errors have been corrected throughout the manuscript.

Reviewer 4 Report

The authors have addressed all my queries.

Author Response

Thanks for your comments